# Building a Production-Ready Multi-Label Classifier for Legal Documents with Digital-Twin-Distiller

**Gergely Márk Csányi** [1], **Renátó Vági** [1,2], **Dániel Nagy** [1], **István Üveges** [1,3], **János Pál Vadász** [1,4], **Andrea Megyeri** [5] **and Tamás Orosz** [1,*]

1   MONTANA Knowledge Management Ltd., H-1097 Budapest, Hungary;
    csanyi.gergely@montana.hu (G.M.C.); vagi.renato@montana.hu (R.V.); nagy.daniel@montana.hu (D.N.);
    uveges.istvan@montana.hu (I.Ü.); vadasz.pal@montana.hu (J.P.V.)
2   Doctoral School of Law, Eötvös Loránd University, Egyetem Square 1-3, H-1053 Budapest, Hungary
3   Doctoral School in Linguistics, University of Szeged, Egyetem Street 2, H-6722 Szeged, Hungary
4   Institute of the Information Society, National University of Public Service, H-1083 Budapest, Hungary
5   Wolters Kluwer Hungary Ltd., Budafoki Way 187-189, H-1117 Budapest, Hungary;
    andrea.megyeri@wolterskluwer.com
*   Correspondence: orosz.tamas@montana.hu

**Abstract:** One of the most time-consuming parts of an attorney's job is finding similar legal cases. Categorization of legal documents by their subject matter can significantly increase the discoverability of digitalized court decisions. This is a multi-label classification problem, where each relatively long text can fit into more than one legal category. The proposed paper shows a solution where this multi-label classification problem is decomposed into more than a hundred binary classification problems. Several approaches have been tested, including different machine-learning and text-augmentation techniques to produce a practically applicable model. The proposed models and the methodologies were encapsulated and deployed as a digital-twin into a production environment. The performance of the created machine learning-based application reaches and could also improve the human-experts performance on this monotonous and labor-intensive task. It could increase the e-discoverability of the documents by about 50%.

**Keywords:** legal document classification; legaltech; digital twin; multi-label documents; multi-label classification

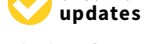



## 1. Introduction

Nowadays, many applications are trying to harness the power of machine learning, and the legal field is no exception; information extraction [1], document classification [2–4], summarization [5,6], are anonymization [7–9] are just a few examples. These applications can significantly facilitate legal practitioners' work delivering faster and/or higher quality solutions. One main source of interest is the decisions of the courts that are (or at least a fraction of them are) usually available free of charge in each country of the European Union [10]. However, finding relevant court decisions is not always an easy task. It is a common problem for lawyers that the different legal databases and legal search engines are both over-inclusive and under-inclusive, especially those which only use keyword search [11]. It is a reoccurring problem that, after a search for court documents, the result list does not contain all the relevant documents, but contains many other documents that practically represent noise. One way to improve the efficiency of legal databases is to categorize legal documents according to some aspects.

When composing legal decisions, the judges tend to summarize the topic or the aim of that decision, which is called the subject matter of the case. The subject matter is a useful piece of information, e.g., in case a lawyer or any other law practitioner is looking for similar cases to the actual one they are working on. However, judges often have

different views on the merits of a case, sometimes too general, sometimes too specific, sometimes highlighting only one particular aspect. As a result of this, the number of used expressions is constantly increasing, reaching tens of thousands of different subject matter versions, currently reaching approximately 28,000 variations in case of court decisions in Hungarian. Hence, a need for reducing the number of subject matters, and a need for a solution that is less dependent on the different points of view or writing styles of different judges, appeared.

By adding labels of the subject matters to court decisions, the efficiency of legal databases can be improved due to the enrichment of the database by new metadata. As the labels can be used to filter the whole document set to more relevant ones, this facilitates and speeds up legal practitioners' work. Moreover, it can help make better prediction models. Prediction, which means determining the outcome of a lawsuit, is a recurring topic in legal technology, and most of the studies treat it as a simple categorization problem [12–16].

Categorizing court decisions, however, is not an easy task. Currently, there is no labeled dataset available for performing subject matter classification. Annotating these relatively long legal documents needs legal editorial experience and is a monotonous, time-consuming job. Using active learning methods can be profitable, but it had to be discarded due to the significant human effort involved, as a legal expert can label around 15–20 documents in an hour. During the project, there was not enough time to generate the right amount of quality training data and then manually perform further tagging for more than 200 different models using active learning. Labeling 100 randomly selected documents took eight hours of work for two independent annotators, respectively. The labeling of the proposed dataset would take about eight years if completed manually. Categorization of legal documents belongs to the multi-label classification problem category [17–20], where one document can be characterized by multiple subject matters at the same time. This especially stands for the documents from the criminal law area where there is usually more than one crime involved in a single case, e.g., someone can be accused of committing murder and theft and robbery at the same time.

Jurisprudence documents show two major differences compared to openly available datasets: one is the average length of the documents and the other is the domain-specific language. The length of a legal case is usually significantly longer than the documents used for comparison in the case of document classification tasks [3], e.g., the Reuters-21578 dataset [21] or the IMDB dataset [22]. Figure 1 shows the distribution of our documents in terms of document length. The lengths were calculated by splitting the documents by whitespace characters after filtering empty strings, so the punctuation characters were not counted separately. Table 1 shows the differences between the above-mentioned datasets [21,22] and our dataset by comparing the number of classes, the document counts, and lengths of the documents. Although the IMDB and Reuters datasets are comparable to ours in the number of classes and the number of documents, respectively, there is an order of magnitude difference considering the average word counts.

Another difference from the previous examples is that, in the case of the subject matter classification, one document can have more than one label. There are many solutions published in the literature to resolve the legal text classification problem. In [3,4,23], the authors resolved this problem as a multi-class classification problem, not a multi-labeling problem [17,18]. Wan et al. proposed a solution for classifying long legal documents [3] by means of a bidirectional Long-Short Term neural network (Bi-LSTM) based solution. However, their task was a multi-class and not a multi-label classification problem. A recent study conducted a multi-label legal document classification task on procedural postures by proposing a deep learning-based solution using label-attention and domain-specific pre-training [2] on an 50,000 document based corpus. Katz et al. [12] built a time-evolving random forest classifier to predict the behavior of the Supreme Court of the United States using more than 240,000 justice votes and 28,000 cases outcomes over nearly two centuries and achieved an accuracy of 70.9%. To predict the decisions of the European Court of Human Rights, Aletras et al. [13] created a binary classification task where the input was

the textual content extracted from the cases, and the output was the judgment of the court whether there had been a violation of an article of the convention of human rights. Li et al. [14] formalized the prison term prediction as a regression problem, adopting a linear regression model and the neural network model to train a prison term predictor based on judgment-specific case features from textual fact description for criminal cases. Bambroo and Awasthi [24] published a DistilBERT-based solution for categorizing more than 300,000 U.S. legal documents by introducing global attention. Nevertheless, their corpus was already labeled, and they also performed multi-class classification, giving only one label to a specific document. Probably the most similar study to ours has been presented by Chen et al. [25] where they performed legal text classification on 30,000 U.S. legal documents, putting them into 50 different categories. They reached the best performance with domain-concept-based [26] Random Forest classifiers that outperformed deep learning based solutions (Bi-LSTM, CNN) in a wide variety of text representation forms (such as GloVe [27], Word2Vec [28], and BERT [29]) even when attention mechanism [30] was also used. However, their corpus was already labeled, and they also performed multi-class, not multi-label, classification.

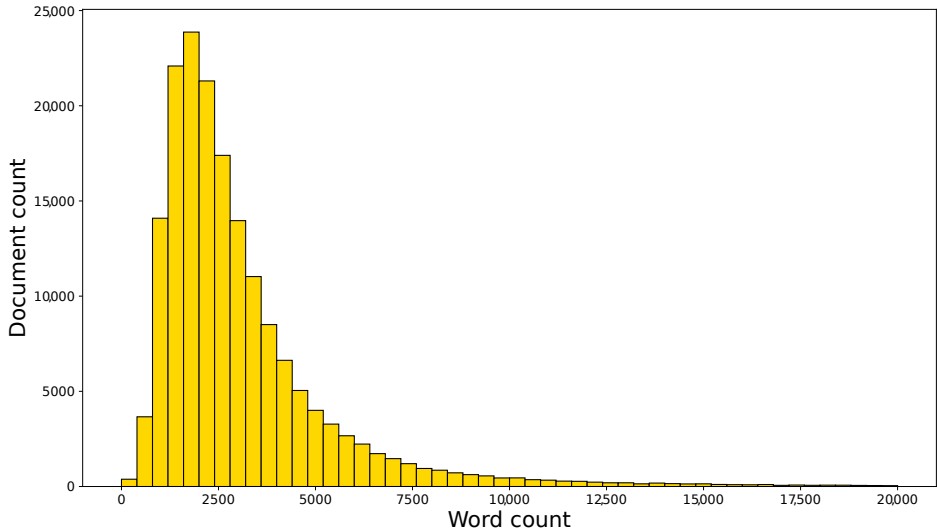

**Figure 1.** The distribution of court document lengths in words.

**Table 1.** Differences between datasets for document classification.

| Dataset | Nr. of Classes | Nr. of Docs | Avg. Word Count Per Document |
|---|---|---|---|
| IMDB | 90 | 10,789 | 144 |
| Reuters | 10 | 135,669 | 393 |
| Subject matter | 167 | 173,892 | 3354 |

The paper aims to label a big dataset composed of lengthy legal cases. The proposed algorithm was integrated into the `digital-twin-distiller`, the created model was saved as a digital twin, and its encapsulated version is deployed in the production environment. The results show the application reaches human-level performance; moreover, using the results, the human performance could be also improved [31]. The paper describes the applied methodologies and the performance comparison of the different machine-learning methodologies on classifying long legal documents. The source code of the proposed algorithms is accessible from the `digital-twin-distiller` projects repository (https://github.com/montana-knowledge-management/digital-twin-distiller, accessed on 25 January 2021). The paper is organized as follows. Section 2 presents the Materials and Methods describing the dataset and the labeling process. In Section 3, the applied and tested methodologies are presented about the vectorization of the documents. In Section 4,

the results of the study are presented; these are discussed in Section 5. Finally, we conclude in Section 6.

## 2. Materials and Methods

### 2.1. Dataset

During this study, a legal dataset was used containing 173,892 legal decisions in Hungarian language provided by Wolters Kluwer Hungary. Every decision can be categorized into five law areas, namely: administrative law, civil law, criminal law, economic law, and labor law. These categories were provided for each document by Wolters Kluwer. From the whole dataset, 169,374 documents were used during the training phase and 4518 documents were held back as a test set. It is important to point out that neither the training nor the test set was labeled beforehand. Hence, the test set was labeled manually to perform the evaluation. The number of documents belonging to the different law areas can be seen in Table 2. The documents where the law area was not specified were not used during training.

**Table 2.** Number of documents belonging to different law areas.

| Law Area | Administrative | Civil | Criminal | Economic | Labor | Not Stated |
|---|---|---|---|---|---|---|
| Training set | 30,891 | 72,525 | 29,751 | 21,125 | 15,063 | 19 |
| Test set | 1207 | 1602 | 591 | 729 | 369 | 20 |

### 2.2. Labeling Process

A labeled dataset is required to apply the supervised machine learning method. However, the dataset originally received from Wolters Kluwer did not contain any labels, except for certain parts of the text extracted by regular expressions referring to the subject matter of the case. These subject matters generally appear at the beginning of the legal documents and are composed by various judges, so the same subject matter might be phrased differently. Therefore, two court cases dealing with very similar issues did not necessarily contain the same text on the subject matter. In many cases, these subject matters extracted from the documents are misleading: they can be too broad (e.g., "administrative case") or too specific (e.g., "payment of X amount of money") in meaning, or just incomplete by mentioning only one aspect of the case (e.g., "committing fraud and other crimes"). At the beginning of the study, the subject matters extracted from the document were available as a list containing approximately 28,000 expressions. Therefore, the subject matter extracted from the text was not sufficient for performing a subject matter classification.

Hence, we have created a legal category system consisting of 167 elements by unifying the extracted subject matters and introducing further generalizations. Two aspects had to be taken into account during the determination of subject matter elements: on the one hand, the various categories and their names needed to comply with the rules of Hungarian legal dogmatics. On the other hand, each class needed to contain a minimum number of documents as training data. A common problem in supervised machine learning classification of legal documents is that, often, there is not enough training data to create an effective categorization algorithm [32].

Positive samples were chosen using a rule-based method by selecting the most appropriate elements from the list of extracted subject matters. The documents that could not be assigned to a label were handled as completely unlabeled data and were not used during the training process.

## 3. Applied Methodologies

### 3.1. Vectorization

The main focus was on finding expressions while keeping the solution simple yet effective. To tackle the problem, the vectorization process Term-Frequency Inverse Docu-

ment Frequency (TF-IDF) vectorization was chosen [33,34]. By using TF-IDF vectorization, simple vectors can be found for documents. The major drawback of this vectorization is that this handles the documents as a bag-of-words, without keeping word order information. However, by using n-grams ($n > 1$) during vectorization, multi-word expressions can be learned from the text. Legal cases usually contain key expressions that can consist of only one word or multiple ones. Therefore, during this study, 1 and 2-g (uni- and bigrams) were used during vectorization, so single words and word pairs to be able to identify both types of key expressions more efficiently.

### 3.2. Preprocessing

As the Hungarian language is an agglutinative language [35] and reducing dimensions is important when using TF-IDF vectorization, the following preprocessing steps were applied: stemming, using `hungarian-stemmer` (https://github.com/montana-knowledge-management/hungarian-stemmer, accessed on 25 January 2021) for Hungarian [36,37], punctuation filtering, number filtering, and lower-casing words.

From the texts, law references were extracted and normalized by using a regular expression-based solution of Montana Knowledge Management Ltd. The law reference extractor returns a list of the law references found in the legal document in the most specific form possible. For instance, when the Act CLXI of 2011 is referenced with or without mentioning a section, only the one with a section is extracted. The same applies to subsections as well. The generalized format of a law reference is: <name of the act>.<section>.§.<subsection>. In order to provide meaningful features for training, the extracted law references had to be broadened. This broadening was performed by the operations as follows: ["<name of the act>.<section>.§.<subsection>"] to ["<name of the act>.<section>.§.<subsection>", "<name of the act>.<section>.§", "<name of the act>"] and similarly ["<name of the act>.<section>.§"] to ["<name of the act>.<section>.§", "<name of the act>"].

### 3.3. Dimension Reduction

As, by default, all words appearing in the corpus form a dimension of the document vector, the dimension of these vectors is very high (millions, in our case); meanwhile, only a tiny subset of the dimensions are essential to perform the classification. Moreover, keeping non-necessary words/n-grams of the vectorization model is sub-optimal from the perspective of the model's size and speed of vectorization.

The Analysis of Variance (ANOVA) algorithm [38] was used to meaningfully reduce the sizes of the TF-IDF vectors selecting K number of best features from the result of the ANOVA algorithm as we wanted to keep the features that correlate best with the labels and keeping the model size low. Nevertheless, ANOVA has been shown to reach similar performance compared to the Principal Component Analysis (PCA) in [39]. Selecting the K best features can be dangerous when the $K$th feature is one of a subset where the features have the same coefficients, as the features selected in this case may differ after each execution.

Generally, there are about 100 words or word pairs which characterize the given subject matter [25,40]. However, we experienced that the size of the training samples affected the quality of the features after feature reduction. The more training data available, the better the reduced features were. We tested the following feature sizes: 500, 1000, 2000, 5000, and 20,000. We found $K = 20,000$ setting as an optimal trade-off for maximizing the likelihood of retaining valuable features yet effectively reducing vector dimensions.

### 3.4. Negative Filtering

Both positive and negative samples are needed to perform traditional supervised training on a binary dataset. Initially, all documents that did not belong to the positive samples were considered negative. Negative samples were drawn only from the same

law area to which the given document belonged. The multi-label classification task was completed by training binary classifiers for each label as in [41–43].

Initially, only the positive training set was known, and the negative samples were the documents from the same law area that did not belong to the same subject matter as the chosen one. However, it is not ensured that all of the positive samples have been labeled during the labeling. Therefore, the negative samples could contain positive samples as well. In this project, to tackle this issue, negative filtering was used. Negative filtering means excluding documents from the negative training set based on legal expressions or law references that are undoubtedly connected to the selected subject matter (Figure 2). The technique itself is related to defining an anchor set, so to define a subset of the instance space to be positive by partial attribute assignment [44–46], but in case of negative filtering, the defined attributes are used to reduce the size of the negative dataset. The filtered documents are included neither in the positive nor the negative samples during training.

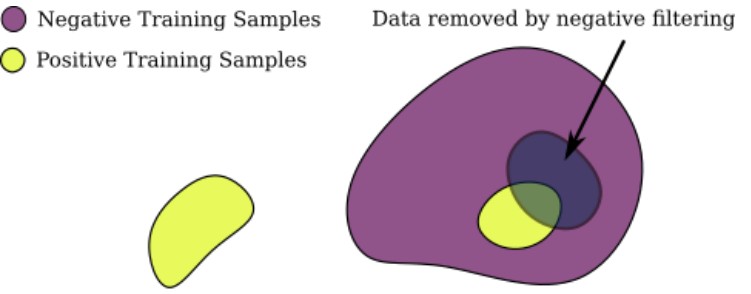

**Figure 2.** Negative filtering to reduce positive label contamination in the negative training set.

Ideally, the positive (yellow-colored) set should be excluded by negative filtering from the negative training data. However, as shown in Figure 2, it is more likely that only a subset of the possible positive samples can be filtered. It is clear that negative filtering can only be used with careful attention, e.g., expressions widely used across different subject matters should not be excluded from the negative training set as, in that case, the machine learning model would learn these expressions. One could argue that the only expected result is that the classifiers will learn the expressions filtered from the negative set by negative filtering. While this statement can certainly be true, and partly the point of this process, an additional benefit comes from this process. This is because filtering the positive samples from the negative ones will make the training sets more accurate. Hence, it is more likely that relevant expressions will remain after feature selection, and if there are, e.g., two closely related legal terms, and only one of them is filtered, the other one is likely to be found and used during training.

Many law references can be strictly connected to specific subject matters. Filtering by these was used during the study.

*3.5. Training*

During training, separate models were trained for the same subject matters that appear in multiple law areas, resulting in 229 different binary classification models while we had only 167 different subject matter categories. The project's goal was to assign a maximum of four labels to each document from the 167 subject matter categories. In many tasks, the transformer-based solutions (e.g., BERT [29]) or deep learning-based solutions (e.g., [47]) usually provide superior performance compared to traditional machine learning methods. In case of document classification, Support Vector Machines (SVM), Logistic Regression (LR) classifiers can provide simple yet satisfactory solutions for long legal documents [3,48–51]. We opted for traditional machine learning solutions for two reasons. Firstly, a very similar study to ours has shown that, in legal document classification, deep learning methods can be overperformed by traditional machine learning classifiers [25]. Secondly, we had only

two months for the whole project and more than 200 models to train. Traditional machine learning algorithms typically learn much faster than deep learning-based ones [52].

Five machine-learning algorithms were tested and evaluated: SVM with linear kernel, Naive Bayes (NB), Logistic Regression, Nearest neighbors (NN), and Random Forest (RF). Where possible, the balanced class weight setting was used, helping the classifiers to reach better performance on the imbalanced datasets (the positive negative ratio varied between 1:2.2 and 1:2416). We evaluated the models performing two-times-repeated 10-fold cross-validation on each training set. During the cross-validation, the folds were selected in a stratified manner. As an evaluation metric, $F_1$ score for the True label was used. The performance of the classifiers were compared on the "Termination of employment" subject matter on texts after the following preprocessing steps: stemming, punctuation filtering but without legal reference extraction, and number filtering (see Figure 3). This subject matter was selected because we could find enough training data, more than 5000 positively labeled documents. As negatives, all other documents were used that could be labeled as belonging to other categories from the labor law area. Due to the lack of time during the project, originally all the classifiers were trained with default hyperparameter settings, except for setting `class_weight=''balanced''` due to the imbalanced nature of the dataset. Hyperparameter tuning was only performed on classifiers not meeting the minimum criteria (having around 80% $F_1$ score after cross validation).

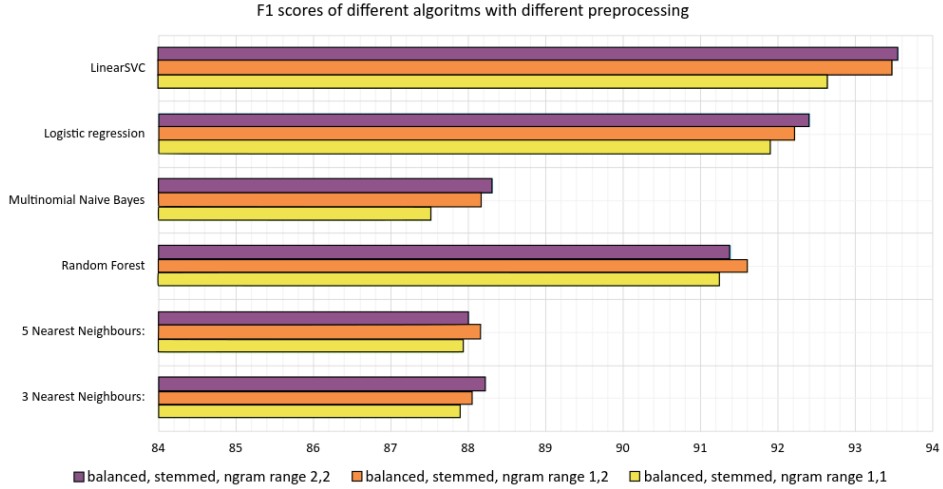

**Figure 3.** Performance of different classifiers on "Termination of employment" subject matter from labor law area, and effect of n-gram range.

The linear kernel SVM proved to be the best algorithm among the tested machine learning methods, reaching an $F_1$ score of around 93.5%. Hence, this method was used as the default classifier for all subject matters. The advantage of using 2-g (word pairs) as features was clear in all cases: all classifiers provided a superior performance when using bigrams compared to the unigram vectorizing solution. We attempted to identify phrases by gensim [53];, the collocation technique was tested, but we did not manage to achieve meaningful results.

### 3.6. Iterative Labeling

Once the models have been trained, possible positive samples were selected and validated by a legal expert. Then, the correctly labeled documents were added to the positive samples in the training set and, if needed, the whole process was repeated. These iterations were time consuming, because every occasion required human labor as well. For this reason, only categories with 20–100 samples were updated this way.

## 4. Results

### 4.1. Determining the Minimum Number of Documents Needed for Classification

After the rule-based labeling of documents, a relatively big category, "Termination of employment", containing around 5000 documents, was selected to choose the labeling methodology, but the minimum document count was needed for training. Positive samples were selected randomly, following a logarithmic scale in counts: 10, 20, 50, 100, 200, and 500. The negative samples were selected from the labor area law documents, that were not labeled as "Termination of employment" or as "Other". The linear kernel SVM model was trained and evaluated on the rest of the labeled documents on the selected positive and negative samples. At each step, the random selection of the documents was repeated ten times, and the results were averaged.

Figure 4 shows the results of the classifiers after punctuation filtering, stemming, and using 2-g for vectorization for different positive sample counts, selecting the number of negatives in a stratified manner.

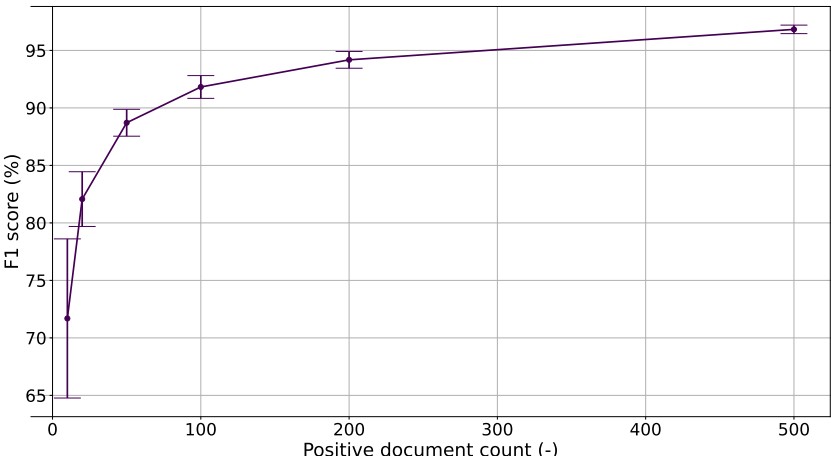

**Figure 4.** Effect of positive document counts on classifier performance

We experienced a cut-off point around 50 positive samples, reaching almost 90% $F_1$ score however, 20 positive samples were enough to reach 80% $F_1$ score. Two thirds of the label categories contained more than 50 documents.

### 4.2. Data Augmentation

The dataset contained a significant proportion of categories with a very low abundance. A widely used approach for boosting the classifiers' performance is to use text augmentation, which means increasing the number of documents by generating "synthetic" instances [40,54]. In this study, we have tried using Easy Data Augmentation (EDA) text augmentation techniques which includes Random Deletion, Random Insertion, Synonym Replacement, and Random Swap [40,55]. As the relevant legal expressions usually occur in a specific order in the text, we wanted to preserve these, therefore we did not use the Random Swap technique during this study. EDA augmentation techniques have been successfully applied in sentiment-analysis of short Hungarian political texts to increase the efficiency of classification [36]. The methods are openly accessible as a Python package via pip (https://pypi.org/project/text-augmentation/, accessed on 25 January 2021).

From the original *"Termination of employment "* dataset, different sizes of positive samples were selected from the original ca. 5000 documents to measure the performance of the augmentation. The initial numbers of these sets were: 10, 20, 50, 100, 250, and 500 documents. In this study, these numbers are referred as base numbers because these numbers served as a base for the augmentation. We have compared datasets where the ratio of positive and negative labels was 1:1 and 1:2.57 (the latter is the stratified ratio on the whole dataset). During the experiment, the effect of augmentation was investigated by

augmenting the original data to a step higher in the base number sequence (for example, producing 20 documents from 10 or 250 from 100). Each result was gained by repeating the calculations 10 times and averaging them.

First, we determined how much performance is gained by using data augmentation compared to not using any. This was calculated by subtracting the $F_1$ value measured on the original data from the $F_1$ value measured on the augmented data. For instance, if we augmented 10 documents to 20 (10 original and 10 augmented samples), then the $F_1$ score found with this dataset was compared with the $F_1$ score from using just the original 10 documents. The difference between these two metrics is indicated in Figure 5 as $\Delta A$. Next, we considered the difference between the augmented and the real, larger dataset. For this, we subtracted the $F_1$ value measured on augmented data from the $F_1$ value measured on original data. For instance, the $F_1$ score calculated using 20 augmented documents was subtracted from the $F_1$ score measured on 20 original documents. This difference is marked with $\Delta B$.

The augmentation methods have an "Alpha" parameter ($\alpha$), which refers to the ratio of the changed tokens (words) during the augmentation process [40,54,55]. For instance, if $\alpha$ is set to 0.1, then 10% of the tokens will be modified during the augmentation (however, in case of Synonym Replacement, the number of tokens replaced may differ, as this method uses WordNet [56,57] to identify synsets, and if the given token is absent, no change will be made).

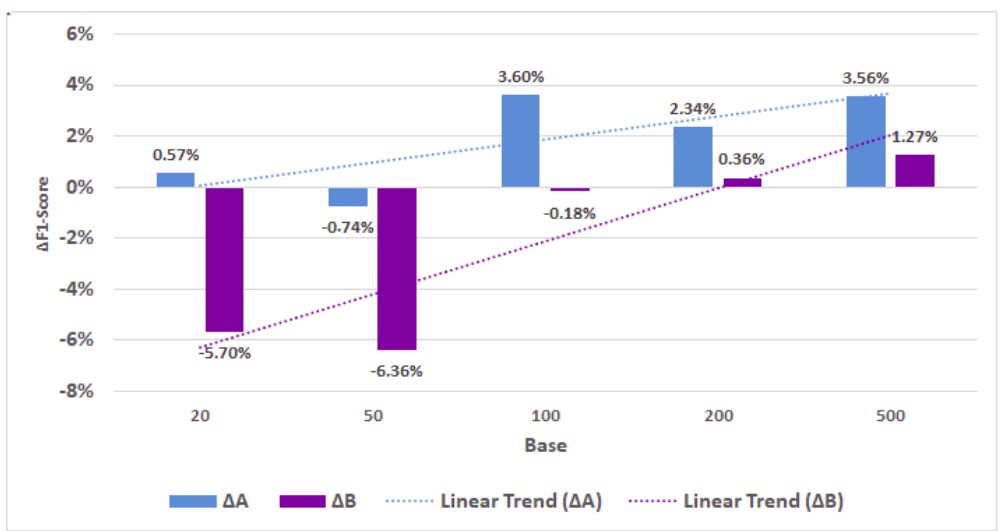

**Figure 5.** $F_1$ score performances at 1:1 positive negative ratio dataset, Alpha 0.1, and Random Deletion method.

As Figure 5 shows, adding augmented data to the dataset improved the performance of the classification in almost every case. The classification performed well with the same size of original and augmented data, suggesting that with relatively small datasets (<100) the original data significantly outperforms the augmented data, but after 100 documents, the results seem to even out.

Figure 6 shows a more detailed overall comparison of results achieved by the applied techniques and parameters. The results show that in the, augmented cases, 100 documents represent some kind of threshold. Below this threshold, $F_1$ scores are greatly varied. At the same time, once this threshold is crossed, the $F_1$ scores are equalized.

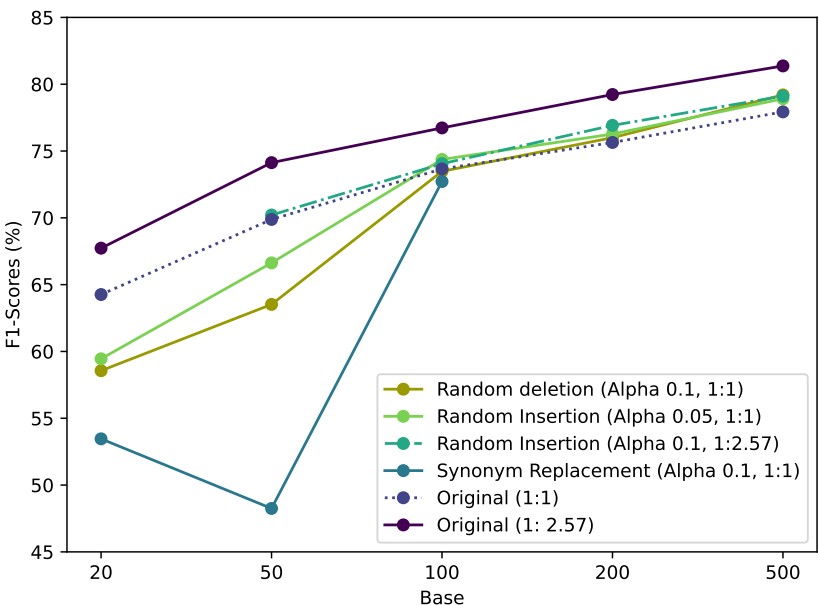

**Figure 6.** Overall comparison of different text augmentation methods with original data.

*4.3. Effect of the Negative Filtering and Addition of Validated Positive Training Data*

Figure 7 shows the effect of negative filtering and the addition of validated training data. All scenarios were evaluated by using linear kernel SVM models from the scikit-learn package [58]. Default settings were used, except for setting the ''`random_state`'' and `class_weight`=''`balanced`'' variables in order to achieve better results due to the imbalanced training scenario. The evaluation was performed in the way presented in Section 3.5, and also keeping the ''`random_state`'' variable fixed to ensure meaningful comparison.

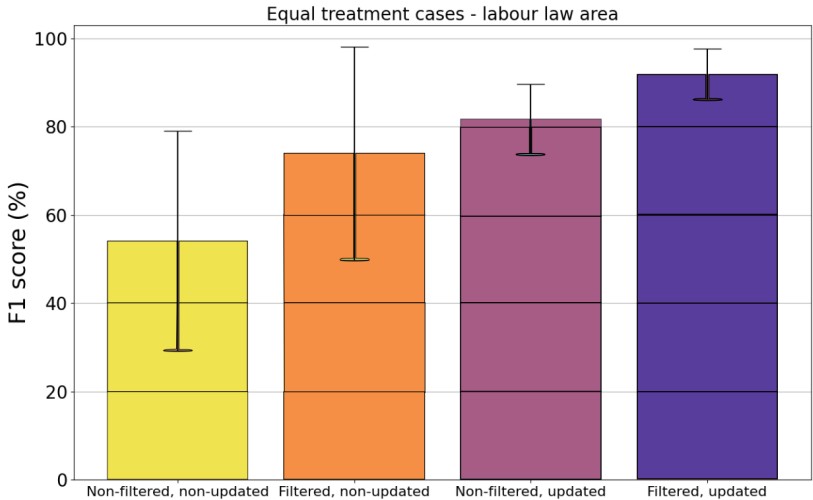

**Figure 7.** Effect of negative filtering and positive training data addition.

Table 3 shows the counts of positive and negative documents during training besides the average, standard deviation, minimum, and maximum $F_1$ scores.

The results show that the addition of validated documents and filtering the negative samples played an important role in improving the classifier's performance. The results suggest that filtering negatives did not affect the standard deviation significantly (reducing from 24.82% to 24.14% and from 7.87% to 5.77%). Nevertheless, it significantly improved the average of the $F_1$ in both cases from 54.15% to 73.95% and from 81.71% to 91.97%, reaching an approximate 20 and 10% increment, respectively. The addition of positive

samples reduced the variance of the $F_1$ significantly (from 24.82% to 7.87% and from 24.14% to 5.77%), but also boosted the average performance by approximately 27% and 18%, respectively (from 54.15% to 81.71%, and from 73.95% to 91.97%).

Using this experience, the low-performance classifiers can reach about 80% $F_1$ score, can perform well for a practical application.

**Table 3.** Number of training samples and $F_1$ scores after cross validation. APD: Added positive validated data, NF: negative filtering.

| APD | NF | Positive Count | Negative Count | Average (%) | Std (%) | Min (%) | Max (%) |
|-----|-----|----------------|----------------|-------------|---------|---------|---------|
| No | No | 32 | 10,012 | 54.15 | 24.82 | 0 | 100 |
| No | Yes | 32 | 9072 | 73.95 | 24.14 | 0 | 100 |
| Yes | No | 102 | 10,012 | 81.71 | 7.87 | 60.87 | 94.74 |
| Yes | Yes | 102 | 9072 | 91.97 | 5.77 | 82.35 | 100 |

*4.4. Validation on Benchmark Set*

A benchmark set was created by selecting 150 documents randomly from the held-out test set. Two experts annotated and cross-checked this dataset to ensure additional attention and reduce the probability of mistakes. After comparing the annotations with the machine-learning-based labeling results, mistakes in seven documents were identified. Two documents were mislabeled and were fixed based on the returned value of the classifier, one document received a label that was not present in the particular area of law, and in an additional four cases, a misspelling-type error occurred. Humans are typically prone to the two latter error types, while machines perform better. It shows the difficulty of this task that the human-level performance is lower than 100% after a cross-check, while the processing time of the total 170,000 documents would take about eight years of work with this methodology.

The results of the benchmark are discussed in Table 4. It can be seen that about 57.3% of the randomly selected documents partly or wholly matched with labels given by the annotators. About 43% of the documents did not have a common label with the annotator-created labels. However, by examining the not matching documents, 22 out of the 64 documents have been given an other-typed label by the classifier, not providing the most specific label, but these are by no means wrongly labeled documents. As these legal categories are not totally disjunct sets (so many of them overlap), more than one solution can be acceptable, while the legal experts only gave the most specific labels to the documents.

**Table 4.** Results on 150 documents tested manually on held-out test set.

| | Did Not Match | Partly Matched | Completely Matched | Sum |
|-----|---------------|----------------|--------------------|-----|
| Count | 64 | 40 | 46 | 150 |
| Percentage (%) | 42.67% | 26.67% | 30.67% | 100% |

Another 14 of the 64 wrongly labeled documents had semantically similar labels to the ones given by the expert. For instance, "Transport and freight contract" and "International transport contract" or "Matters concerning business organizations" and "Matters concerning companies" are labels closely related to each other semantically. When a human legal expert annotates, they take into consideration not only the words in the written text, but also their legal background knowledge. Such categories proved to be hard for the classifiers to identify due to the selected vectorization form. Nevertheless, another 14 documents suffered from wrongly defined labels (defining a too broad category), and the classifier made mistakes in 14 documents. On a document level, 28 out of 150 documents were mislabeled, an 18.67% error rate.

Correlation between Categories

During the validation of the cases, an interesting phenomenon was identified: how the classifier tended to make mistakes. This happened when a legal category did not occur by itself, but always in pair with other labels, for which *Connivance* label is a good example. This originates from the meaning of connivance: someone cannot commit this crime without another crime being committed. However, if the category behaves like this and the joint categories are not uniformly distributed, e.g., in the majority of the cases *Connivance* label is alongside *Fraud*, the classifier will likely be biased and misclassify cases that are only *Fraud* cases to be *Connivance* cases as well. The cause of this problem is that the labels are not independent from each other [59], hence two correlated labels tend to have more common features.

Another source of error was due to the semantic similarity between categories. In these cases the labels might be interchangeable (e.g., *Transport and freight contract* and *International transport contract*).

### 4.5. Experimental Validation

A small experiment was made to measure the human-level performance on the subject-matter labeling task [31]. We have selected 220 documents and annotated them by legal experts as a reference set. The documents were selected in a stratified way. Six legal experts who professionally edit legal texts were selected to annotate these documents within three hours. The task was to annotate as many documents as they could. We thought that it was impossible to label this large amount of documents in the given time. The aim was to simulate real working conditions and to be able to examine the effect of fatigue during labeling that has been shown to be a major drawback of human labeling [60]. The legal experts were divided into two groups: the members of the first group could see the result of the computer annotations and they had to select and complete the valid labels for a document, while the second group had to make the annotation without the computer assistance.

Due to the limited time and the large amount of the documents, we introduced a point system to measure the discoverability and the performance of the different participants. Due to the different sizes of the different label groups we gave different points for a good label: 1, 5, or 10 points. The scores represent the different information content of the different labels. For example, a label such as *Termination of employment*, which represents 30% of the labels of the labor law area represent a relatively low information content, was worth 1 point, while *Medical malpractice*, which is a relatively rare label, was worth 10 points. The reference set contained 1020 points of information.

We obtained surprising results after the experiment as we checked the overall performance of the different participants (see Figure 8). Those editors who could use the assistance of the computer scored only 245 points and had a worse performance than those colleagues who could not use the assistance of the computer (342), and the computer-scored the most, 490 points. However, if we check the average performance on a document, we see quite a different picture. In this kind of comparison, editors assisted by the computer mined over 50% more information from the documents than the other editors and the computer. This was because the editors who could use computer assistance labeled the documents slower, so they managed to label fewer documents during the dedicated three hours. However, as the average points earned in Figure 8 shows, the computer-aided editors were able to add more specific and, therefore, more valuable labels for the documents on average than the ones without computer assistance. These results suggested that those editors who could use the labels given by the computer paid more attention during reading through a document, which resulted in a higher rate of finding rare document labels that worthed more points. Editors who did not use computer aid more often used tags that were more straightforward, with less focus on less frequent tags, resulting in a lower points per document score (see Figure 8).

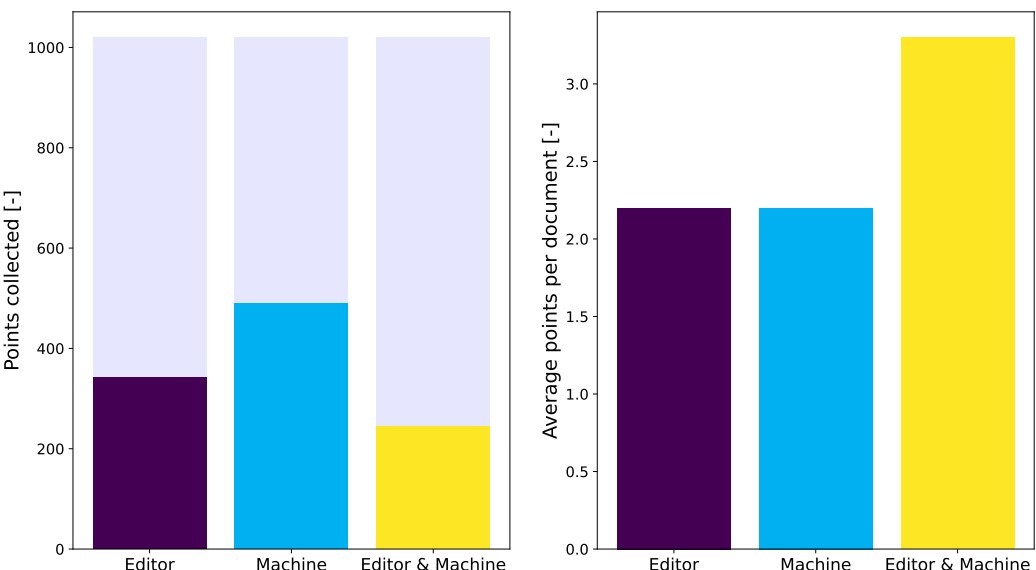

**Figure 8.** Comparison of performances: points collected (**left**) and average points found per document (**right**).

## 5. Discussion

The solution presented here differs from similar previous studies in two significant respects: first, that we performed multi-label classification on the legal data, and second, that we did not initially have a labeled dataset [24,25]. The most challenging labels for the machine learning methods were those where the legal category was not independent of other legal categories. Connivance, is the most illustrative example as this crime never occurs by itself; one cannot commit connivance without another crime being involved. Therefore, this legal category contains many keywords from joint legal categories. Those categories are also problematic for the machine-learning methods, which has semantically similar names and works with semantically similar subject matters such as *Transportation* and *International transportation*.

Different machine-learning techniques have been tested on a selected label to determine and compare the performance of the machine-learning classifiers. The *Termination of the employment* label was selected, where we had more than 500 positive training data. The easy data augmentation (EDA) techniques have been originally developed and tested for short text augmentation [55]. However, our results show that these text augmentation methods can be successfully applied to increase the performance of the binary classifiers on relatively long legal texts. We reached the highest $F_1$ score increment on legal texts when we trained the machine learning classifier on a 50–100 positive sample-sized training set. We also determined the minimum number of the required documents to be used per category using this dataset (the *Termination of the employment* label). We experienced a cut-off point of around 50–100 documents, but 20 documents were enough to meet our minimal criterion of having around 80% of $F_1$ score.

The overall performance of the realized system has been validated on two benchmark sets. The proposed system (at least partly) correctly labeled 57.3% of the proposed documents. One of the well-known drawbacks of this result is interpretability. Due to the proposed system being applied in a production environment, it was more important to know how this result compares to the performance of human experts than comparing the different machine learning-based techniques to each other. A small experiment was designed to answer this question, where the performance of the machine learning methodology was compared on another dataset. The results show that the overall performance of the machine learning model reaches the performance of the top legal experts. The results show that the learning curve of humans is different from the machine learning methodology.

The results of the machine learning methods can significantly increase (by about 50%) the data discoverability if the human experts can use them to improve their results.

## 6. Conclusions

This paper presents a binary classifier-based multi-labeling solution for legal text classification. The proposed study aimed to find the proper subject matter labels for each document from a corpus containing approximately 174,000 documents. More than one hundred different labels were used in the study, and each document could have a maximum of four distinct labels. The frequency of the labels showed a high variance in the examined set. There were labels which belonged to thousands of documents, while about 30% of the labels belonged to no more than fifty documents. The performance of different machine learning models was compared on a selected label, where we could select different group sizes of 10 to 500. Moreover, the performance of the different text augmentation techniques was also examined on this dataset, because these augmentation techniques were tested initially on short texts, while the proposed analysis showed their effectivity on these relatively long texts. The proposed models were encapsulated into the `digital-twin-distiller` framework and were deployed in a containerized (Docker) form. The proposed application can automatically label the incoming documents through a REST API. The performance of the created machine learning-based application reaches the human-level performance on this labeling task. In addition, using the labels given by the computer could increase the human performance as well. A further study could assess the applications of different vectorization forms, the correlation between the proposed labels, and the application of positive-unlabeled learning and weak-label-learning approaches.

**Author Contributions:** Conceptualization, T.O., G.M.C., D.N. and R.V.; methodology, D.N. and R.V.; software, G.M.C., D.N. and T.O.; validation, A.M., G.M.C. and J.P.V.; formal analysis, G.M.C.; investigation, G.M.C.; resources, V.J., G.M.C.; data curation, A.M.; writing—original draft preparation,G.M.C., T.O. and R.V.; writing—review and editing, T.O. and G.M.C., D. N.; visualization, I.Ü., G.M.C. and D.N.; supervision, T.O.; project administration, D.N.; and funding acquisition, D.N. and A.M. All authors have read and agreed to the published version of the manuscript.

**Funding:** Project No. 2020-1.1.2-PIACI-KFI-2020-00049 has been implemented with the support provided from the National Research, Development and Innovation Fund of Hungary, financed under the 2020-1.1.2-PIACI KFI funding scheme.

**Institutional Review Board Statement:** Not applicable.

**Informed Consent Statement:** Not applicable.

**Conflicts of Interest:** The authors declare no conflict of interest.

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
