# Peer review of "Building a Production-Ready Multi-Label Classifier for Legal Documents with Digital-Twin-Distiller"

_applsci, doi:10.3390/app12031470_

Round 1

Reviewer 1 Report

This paper offers an interesting solution for the legal case problem. This paper utilizes artificial intelligence in terms of digital twin and machine learning to help practitioners to optimize their performance. Overall, the paper is well structured and started with a good introduction although some sections need to be improved. The author needs to more clearly describe the stage for each section and provide a better discussion for the result. Some suggestions are presented for clarifying some points in each section that are summarized as follows:

  1. The authors need to emphasize the result of the approach not only reach the human level performance but also improve the human performance
  2. What is the implication for the result need more description in the abstract

Section 1

  1. The sentence in lines 31 – 34 needs citation
  2. The third paragraph inform about the problem about the efficiency and speed of work, it is needed to inform about previous performance
  3. Line 53 inform about domain-specific language, however, I cannot find it in figure 1 and table 1
  4. Figure 1 and Table 1 need clearly describe in the paragraph
  5. Some abbreviations in the table need clearly stated because not all readers will understand it
  6. Paragraph 6 in line 58-67 provide a good explanation for the previous approach for a similar problem, however, need to be enhanced with informing the performance
  7. The last paragraph in the introduction informs about the aim of the paper, so it is no need to inform about problems like time-consuming and labeling. If it needs to highlight this problem, probably can add to the previous paragraph.
  8. It is good to inform the proposed algorithm, however, it is no need to mention the result for previous research. It is more important to enhance the description of the approach without discussing previous research

Section 2

  1. Did the author have the background to determine the five law areas?
  2. It needs to inform about the consideration to divide the dataset become training and testing
  3. Table 2 inform about a number of each area needs clearly inform about Dev.
  4. Also, table 2 inform about the number of documents that are not stated area, how the authors deal with it?
  5. It is difficult to understand linking with the previous section (dataset), the number of documents (169.374), the number of expressions (28.000), and the number of elements (167). It is need clearly described
  6. At the end of section 2.2 inform about the result which mentions 124.000 documents, it is need clearly informed to get this number
  7. At the end of section 2.2 for methodology section needs to provide a solution not discuss the problem. The problem explanation can move to the introduction section

Section 3

  1. Why consider 1 and 2 grams?
  2. Why consider 20.000 as the default value for K?
  3. If previous research informs deep learning provides superior performance, why does this paper consider machine learning?
  4. For evaluation why only consider F1? How about the accuracy, recall, precision?
  5. Line 209 inform about 5000 positively labeled, how about the other document?
  6. Line 223-234, could the authors provide some reference for this phenomenon?

Section 4

  1. Line 266-267 need to inform the stage for each approach
  2. All of the tables need to revise considering the template
  3. Figure 8 need additional information (8a & 8b). It is needed to describe more for figure 8a and 8b. Why do this two have different results of performance?
  4. To sum up, the section, need more explain the result
  5. The section title is result and discussion, however, I cannot find the discussion section

Section 5

  1. The author needs to consider the objective for this study which not only builds some approach to make it equal with human performance but also can enhance human performance

Author Response

Dear Reviewer,

 Thank you for the time you spent providing a detailed review and please find our answers to your questions in the attached pdf.

Reviewer 2 Report

Congratulations on submitting the manuscript to the MDPI Applied Science journal. Suggestions that could improve the manuscript are given below:

1) At the end of the manuscript abstract, it would be good to present a number of values accuracy, or some other measures which show the performance of the authors' research. It would be useful for readers to know from the abstract of the manuscript what has been obtained and how good its performance is.

2) One of the keys of this manuscript is multi-label documents, so it would be good to add it to the keywords list.

3) The analysis of the related works or at least the solutions of kinda similar is missing in the manuscript. Maybe there is no directly related scientific work as the authors do, but they are for sure solutions and analysis of multi-label text data (short text, documents, etc., the specific of analysis quite same). For example, DOI: 10.15388/22-INFOR473, DOI: 10.1109/IACS.2015.7103229, etc. At least the authors could give a small section of related works, to show the scope of this specific area, that they checked, analyzed what is done in this field. In my opinion, it must be done. Some little parts of the analysis are given in the Introduction section, but I highly recommend splitting it into Introduction and Related works.

4) At the end of the Introduction, the structure of the manuscript could be presented as well.

5) The second column of Table 2 is empty, a minor mistake.

6) In Figure 1 the length of the documents is presented, but does not mention in what type of length? Words or tokens? I mean if the dataset has been preprocessed unnecessary information is removed, but if not, each number, point, and comma is counted as a word in this figure? Perhaps we need to give some details.

7) I would say that we need the argument why in Section 3.3. The ANOVA has been chosen. There are many dimensional reduction methods. Does it better fit your data, or maybe it is better in performance, etc.

8) At least I did not find, how the parameters of each classifier have been selected? By default? Usually, hyperparameter optimization is performed in such research, or maybe a grid search with some limits is used. If not, the authors just need to give the argument why not.

9) 3.6.1 the style of the subsections differs from the rest titles. And I am not sure it needs a separate section; there is no 3.6.2 or other.

10) Table 4 missing border.

11) References need to be formatted according to the style of the Applied journal, now some of them do not meet requirements.

Good luck in submitting the manuscript.

Author Response

(The authors gave the same response as above.)

Round 2

Reviewer 1 Report

Dear,

The paper quality improved as per advice. However, there is still room for improvement. Please consider several points below:

  1. Authors need to add citations for explanation in lines 199-204
  2. Figure 8 needs more description why the editors who could use computer assistance label the documents slower and manage fewer documents?
  3. No need to mention a paragraph at lines 470-473 because it hasn't any contribution to your research
  4. The discussion section still has a different font, maybe the authors should check all of the formats of the manuscript
  5. The author should more describe what is the suggestion for problem mentioned in lines 487-489

Author Response

Please, see the attached document.

Reviewer 2 Report

Thank you for taking suggestions into account. 

Some minor mistakes should be fixed:
1) Line 203, K = 2000 better use math style, as was before over the all manuscript.
2) Usually 2-gram is called bigram, but 2-gram is not a mistake.
3) Line 327, space not needed after the bracket"...20 ( getting..."
4) Over the all manuscript the numbers are separated by a point, e. g. 0.5, but in Figure 5 the comma has been used. Someone could think it is a thousand (obvious it is not, or maybe I wrong?). The same style over the all manuscript has to be. 
5) Everywhere was Figure, but Fig 3 appears in line 488.

Good luck with the final submission.

Author Response

Please, see the attached document.
